# Residential flood loss estimated from Bayesian multilevel models

Guilherme S. Mohor[1], Annegret H. Thieken[1], and Oliver Korup[1,2]

[1]Institute of Environmental Science and Geography, University of Potsdam, 14476 Potsdam, Germany
[2]Institute of Geosciences, University of Potsdam, 14476 Potsdam, Germany

**Correspondence:** Guilherme S. Mohor (samprognamoh@uni-potsdam.de)

**Abstract.** Models for the predictions of monetary losses from floods mainly blend data deemed to represent a single flood type and region. Moreover, these approaches largely ignore indicators of preparedness and how predictors may vary between regions and events, challenging the transferability of flood loss models. We use a flood loss database of 1812 German flood-affected households to explore how Bayesian multilevel models can estimate normalised flood damage stratified by event, region, or flood process type. Multilevel models acknowledge natural groups in the data and allow each group to learn from others. We obtain posterior estimates that differ between flood types, with credibly varying influences of water depth, contamination, duration, implementation of property-level precautionary measures, insurance, and previous flood experience; these influences overlap across most events or regions, however. We infer that the underlying damaging processes of distinct flood types deserve further attention. Each reported flood loss and affected region involved mixed flood types, likely explaining the uncertainty in the coefficients. Our results emphasise the need to consider flood types as an important step towards applying flood loss models elsewhere. We argue that failing to do so may unduly generalize the model and systematically bias loss estimations from empirical data.

*Copyright statement.* TEXT

## 1 Introduction

The estimation of flood losses is a key requirement for assessing flood risk and for the evaluation of mitigation strategies like the design of relief funds, structural protection, or insurance design. Yet loss estimation remains challenging, even for direct losses that can be more easily determined than indirect losses (Figueiredo et al., 2018; Vogel et al., 2018; Amadio et al., 2019; Meyer et al., 2013). Numerous methods of inferring flood damage from field or survey data have been tested, if not validated, with varying degrees of success (Gerl et al., 2016; Molinari et al., 2020).

Without standard loss documentation procedures in place, the highly variable losses caused by different flood types (e.g. pluvial, fluvial, coastal) can make loss modelling particularly challenging, especially where data are limited or heterogeneous. This lack of detailed or structured data motivates most modelling studies concerned with flood loss to assign just a single type of flooding to each event (Gerl et al., 2016). Another confounding issue is scale: inventories of flood damage are often aggregated at administrative levels such as municipalities or states (Spekkers et al., 2014; Bernet et al., 2017; Gradeci et al., 2019).

This aggregation masks links between damage and exposure or vulnerability at the property scale (Meyer et al., 2013; Thieken et al., 2016). These unstructured or aggregated data make damage models prone to underfitting, whilst training models with numerous predictors may lead to overfitting, reducing the ability to generalise and transfer to situations where information is unavailable (Meyer et al., 2013; Gelman et al., 2014; Gerl et al., 2016). Previous work has emphasised this challenge of transferring models with respect to different flood types, events, or locations (Jongman et al., 2012; Cammerer et al., 2013;

Schröter et al., 2014; Figueiredo et al., 2018).

    In this context, multilevel or hierarchic models are one alternative and offer a compromise between a single pooled model fitted to all data and many different models fitted to subsets of the data sharing a particular attribute or group. Bayesian multilevel models use conditional probability as a basis for learning the model parameters from a weighted compromise between the likelihood of the data being generated by the model and some prior knowledge of the model parameters. These models

explicitly account for uncertainty in data, low or imbalanced sample size, and variability of model parameters across different groups (Gelman et al., 2014; McElreath, 2016). There are several approaches to the bias-variance trade-off (McElreath, 2020). We conduct a variable selection through cross-validation to achieve a balance between predictive accuracy and generalization. Using priors in the Bayesian framework is using regularization by design and keeps the model from overfitting the data (McElreath, 2020).

In contrast to empirical models, synthetic models are developed based on expert opinion and offer a good approach to harmonize loss estimations. However, how these models rely on assumptions is problematic when preparedness and other behavioural variables are concerned. In general, synthetic models tend to reduce the variability of data and remain rarely validated (Sairam et al., 2020). Therefore, we train our Bayesian model using reported data.

    In this study, we use survey data from households affected by large floods throughout Germany between 2002 and 2013

(Thieken et al., 2017). These data go beyond addressing physical inundation characteristics by offering a broad view of the damaging process including the flood types that affected the households (i.e., floods from levee breaches, riverine floods, surface water floods, or rising groundwater floods).

    Mohor et al. (2020) used this database to explore the most relevant factors for estimating relative loss of residential buildings with a regression model. From a larger pool of candidate variables, the authors selected 13 predictors of the flood hazard, build-

ing characteristics, and preparedness, including flood type as an indicator, and suggested that the influencing factors contribute with different magnitudes across flood types. Vogel et al. (2018) trained Bayesian Networks and Markov Blankets (MBs) for different flood events and types in Germany, obtaining varying compositions of meaningful predictors. Bayesian Networks focus on the dependence between variables and flow of information (Vogel et al., 2018), rather than the weight of each factor into the final loss, which is the case of Bayesian Inference.

Here we expand on the model of Mohor et al. (2020) by acknowledging structure in the dataset and explore whether a single regression model can apply not only to different flood types, but also to regions or flooding events. Single flood events can affect cities differently across regions, likely reflecting socioeconomic and geographic conditions and building codes, for example. These characteristics reflect a given asset's resistance to the hazard process (Thieken et al., 2005). These characteristics may differ on the level of administrative regions, and hence we considered a multi-level model variant structured by regions.

Additionally, flood preparedness evolved over time, documented, for example, by Kienzler et al. (2015) and Thieken et al. (2016) for Germany. Economic situations may also change the relative value of exposed assets and its recover or repair costs (Penning-Rowsell, 2005; Kron, 2005). Such changes are challenging to include in loss models, however. Therefore, we considered a third model variant structured by flood events, capturing the timely aspect. Therefore, estimate relative flood losses in Germany with a Bayesian multilevel model featuring three different groups, i.e. (i) flood types, (ii) administrative regions, and

(iii) individual flood events to learn which predictors might aid the transferability of loss models. We hypothesise that the effect of some predictors varies with flood type, administrative region, or flood event. We use multilevel linear regression to explore these possible differences. Judging from previous work, we expect differing socioeconomic conditions or preparedness across regions of Germany (Thieken et al., 2007; Kienzler et al., 2015), a gradual development of building standards and preparedness (Kienzler et al., 2015; Vogel et al., 2018), and differing hazard characteristics and resistance across flood types (Mohor et al.,

2020).

## 2 Data and Methods

### 2.1 Data

In this study we use the data from a joint effort that conducted surveys among households affected by large floods throughout Germany to investigate various aspects of the flood damaging process more systematically. Beginning with the large central-

75 European floods of 2002, this database has more than 4000 entries from six different flood events (Thieken et al., 2017). The surveys had approximately 180 questions, with slight adaptations and improvements in clarity in each edition, and were conducted after major floods that hit Germany in 2002, 2005, 2006, 2010, 2011, and 2013. These floods happened in different seasons and involved different weather conditions that led to varying flood dynamics, i.e. riverine floods, surface water floods, rising groundwater floods, and levee breaches (Kienzler et al., 2015; Thieken et al., 2016). While the floods in 2002, 2005,

and 2010 evolved quickly, the floods in 2006, 2011 and 2013 were slow-onset events. In all cases, the eastern and the southern parts of Germany were affected the most.

These data go beyond addressing physical inundation characteristics, and also include aspects of warning, preparedness and precaution at the level of individual households. This gathering of socioeconomic information and building characteristics thus offers a broad view of the damaging process rarely found elsewhere (Thieken et al., 2017). This dataset also specifies the

85 flood types that affected the households in four categories: floods from levee breaches, riverine floods, surface water floods, or rising groundwater floods. Multiple flood types were reported for the same event, even within the same city, thus giving rise to compound events that can be defined as the synchronous or sequential occurrence of multiple hazards (Zscheischler et al., 2020).

From this dataset, Mohor et al. (2020) identified thirteen predictors via variable selection in a multiple linear regression frame-

90 work. Flood type was considered as a categorical or indicator variable (Gelman and Hill, 2007). These selected predictors are ranked in order of importance, according to the number of times the predictor was kept in an iterative variable selection procedure with random sampling (Table 1). A more detailed description of the variables and the method can be found in Vogel

et al. (2018) and Mohor et al. (2020).

In this study, we used three characteristics to group our data: (i) flood type, with categories levee breaches, riverine, surface, and groundwater floods; (ii) regions of Germany, with categories south (Bavaria and Baden-Wurttemberg), east (Brandenburg, Mecklenburg-Western Pomerania, Saxony, Saxony-Anhalt, and Thuringia); as well as west and north (Hesse, Lower Saxony, North Rhine-Westphalia, Rhineland Palatinate, and Schleswig-Holstein – grouped together due to the low number of cases); and (iii) flood year, i.e. 2002, 2005, 2006, 2010, 2011, and 2013. We tested three model variants, each using only one group variable at a time (Table 2). We refer to these model variants as the flood-type model, the regional model, and the event model, respectively.

**Table 1.** Description of potential predictors of flood loss

| | Predictor | abbr. | Unit /description |
|---|---|---|---|
| 1 | Water depth | WD | in cm |
| 2 | Building area | BA | originally in $m^2$; due to high skewness, the variable is log-transformed |
| 3 | Contamination | Con | indicator from 0 (none) to 2 (heavy contamination) |
| 4 | Duration | Dur | originally in h; due to high skewness, the variable is log-transformed |
| 5 | Property-level Precautionary Measures (PLPM) | Pre | indicator from 0 (none) to 2 (very good precaution) |
| 6 | Insured | Ins | yes/no |
| 7 | Perceived efficacy of PLPMs | Eff | Likert-type scale from 1 (highly effective) to 6 (highly ineffective) |
| 8 | Emergency measures | Eme | indicator from 0 (no emergency measures performed) to 17 (many emergency measures performed effectively; Thieken et al. 2005) |
| 9 | Cellar | Cel | yes/no |
| 10 | Relative flow velocity | Vel | Likert-type scale from 0 (no flow) to 6 (very high velocity) |
| 11 | Flood experience | Exp | 5 classes from 0 (no previous flooding) to 4 (more often and recent previous flooding) |
| 12 | Building quality | BQ | Likert-type scale from 1 (very high quality) to 6 (very low quality) |

## 2.2 Methods

Single-level multiple linear regression is adequate for capturing general trends in data, but ignores structure in the data, such as flood type or region affected. We explore the suitability of a Bayesian multilevel model to estimate relative building loss

**Table 2.** Number of instances in the training set used across grouping variables flood type, region, and event year ($n = 1269$)

| Flood Types | Levee Breach | Riverine | Surface | Groundwater | Sum (n) |
|---|---|---|---|---|---|
| Flood events | | | | | |
| 2002 | 110 | 252 | 103 | 106 | 571 |
| 2005 | 8 | 35 | 7 | 6 | 56 |
| 2006 | 0 | 25 | 2 | 3 | 30 |
| 2010 | 31 | 86 | 19 | 5 | 141 |
| 2011 | 1 | 49 | 5 | 11 | 66 |
| 2013 | 108 | 236 | 16 | 45 | 405 |
| Regions of Germany | | | | | |
| South | 52 | 174 | 53 | 58 | 337 |
| East | 205 | 469 | 80 | 111 | 865 |
| West and North (W+N) | 1 | 40 | 19 | 7 | 67 |
| Sum (n) | 258 | 683 | 152 | 176 | 1269 |

(or loss ratio) from models with different predictor combinations. We use a numerical sampling scheme for Bayesian analysis implemented in the `brms` package (version 2.11.1; Bürkner (2018)) in the `R` programming environment (version 4.0.1; R Core Team (2020)). We test and compare various multilevel models with differing complexity. We trained the model on 70% of the complete dataset (no missing data), with a total of 1269 data points in the training dataset and 543 data points in the testing dataset. Although the dataset consists of more than 4000 datapoints, due to random missing data, the testing and training subsets size depends on the variables included in the model. Thus, 1812 datapoints were available in our case.

### 2.2.1 Bayesian multilevel model

Bayesian multilevel models weigh the likelihood of observing the given data under the specified model parameters by prior knowledge. Bayesian models thus express the uncertainty in both the prior parameter knowledge and the posterior parameter estimates. The multilevel approach allows us to analyse all data in one model while honouring structure or nominal groups in the data. Thus, the training of the group-specific parameters occurs at the same time so that model parameters can inform each other by means of specified (hyper-)prior distributions. This approach warrants more training data than running stand-alone models on subsets of our data, which in turn are more prone to over- and underfitting and overestimates of the regression coefficients, while reducing effects of collinearity, and offering a natural form of penalised regression (McElreath, 2016). The (unnormalized) posterior density, i.e. the probability distribution of the model parameter(s) $\theta$ given the observed data $y$ of a Bayesian model is proportional to the product of the prior of the model parameters—a probability distribution describing

previous knowledge about the model parameters—and the plausibility of observing the data given the model under these parameter choices, also known as likelihood (Gelman et al., 2014):

$$p(\theta|y) \propto p(\theta)p(y|\theta) \tag{1}$$

In a multilevel model, the data are structured into $J$ groups, with model parameters allowed to vary between these groups ($\theta_j$). The vector of group-level parameters $\theta_j$ is itself drawn from a distribution specified by hyperparameter(s) $\tau$. The model returns parameter estimates for both the entire (pooled) data and its $J$ groups, although all parameters are learned jointly via the specified distribution of the hyperparameters. The group-level (hyper-)parameters are unknown and learned from the data to inform the posterior distribution. This relationship can be written as the joint prior distribution (Gelman et al., 2014):

$$p(\theta,\tau) \propto p(\tau)p(\theta|\tau) \tag{2}$$

The joint posterior distribution can then be written as (Gelman et al., 2014):

$$p(\theta,\tau|y) \propto p(\theta,\tau)p(y|\theta) \tag{3}$$

The `brms` package is an interface for building multilevel models (Bürkner, 2018) and calls STAN, a programming language for Bayesian statistical inference (Carpenter et al., 2017). STAN uses a Hamiltonian Monte Carlo (HMC) method, a type of random sampling to approximate posterior distributions that are without analytical solutions (Kruschke, 2014), or the extension of HMC, the No-U-Turn Sampler (NUTS), which is the default option in `brms` (Bürkner, 2018).

The choice of the likelihood and the priors should follow assumptions about the data-generation process (Gabry et al., 2019). Our response variable is relative loss, and relates total direct, tangible flood loss such as repair and replacement costs (Merz et al., 2010) to the total asset value of a given residential building; relative loss thus varies from 0 to 1. Recent work on flood loss modelling used an inflated beta distribution to first model the probability of no loss (Rözer et al., 2019), or of total loss using a zero-and-one inflated beta distribution (Fuchs et al., 2019); a beta distribution then serves to estimate intermediate losses (Evans et al., 2000). This approach is useful in cases where flood damages remain unreported or unaccounted for. Our dataset of affected households has only 15 instances where relative flood loss was either 0 or 1. Hence, we dismissed those instances and modelled only partial loss ratios using the beta distribution:

$$y \sim \text{Beta}\left(\mu\phi, (1-\mu)\phi\right) \tag{4}$$

Where $y$ is the loss ratio that we assume follows a beta distribution with parameters mean $\mu$ and precision $\phi$. The mean ($\mu$) is estimated from a multiple linear regression with $K$ predictors as:

$$\text{logit}(\mu_i) = \alpha_0 + \alpha_{j[i]} + \mathbf{X}_{i,k}\beta_{k,j[i]} \tag{5}$$

Where subscript $i$ refers to each datapoint, subscript $k$ refers to the predictors; subscript $j$ refers to the groups; $\alpha_0$ is the population-level intercept, $\alpha_j$ is the vector of group-level intercepts; $\mathbf{X}_{i,k}$ is the $i \times k$ matrix of predictor values; and $\beta_{k,j}$ is the $k \times j$ coefficient matrix. Each data point $i$ is thus a vector of group-level coefficients, expressed by the $j[i]th$-column of $\beta$. The model therefore has one population-level parameter ($\alpha_0$) and $(k+1) * j$ group-level parameters ($\alpha_j$ and $\beta_{k,j}$).

In `brms`, the multilevel structure of the regression specifies Gaussian prior distributions for the intercepts $\alpha_j$ and for the predictor coefficients $\beta_j$ with fixed zero means and unknown standard deviations. The group-level standard deviations are hyperparameters that are common to all group levels, but individual for the intercept or for each given predictor ($\sigma_\alpha$ and $\sigma_{\beta_k}$). Therefore, we use standardised input data that are centred at zero and scaled to unit standard deviation. The prior of each group-level standard deviation is in turn a weakly informative Gamma distribution with shape and inverse scale (or rate) parameters (2, 5), which accumulates most probability mass at low positive values below 1. This choice of prior is appropriate for standardised input data even without any specific prior knowledge, for example, from other studies on flood damage. While previous studies have indicated consistently that the effect of water depth is positive, we decided to keep the priors weak enough to allow for the possibility of either positive or negative estimates for all predictor coefficients to explore possible effects of the multi-level model. The prior for $\phi$ is non-informative.

$$\alpha_j \sim \mathcal{N}(0, \sigma_\alpha) \tag{6}$$

$$\sigma_\alpha \sim \mathrm{Gamma}(2, 5) \tag{7}$$

$$\beta_{k,j} \sim \mathcal{N}(0, \sigma_{\beta_k}) \tag{8}$$

$$\sigma_{\beta_\mathbf{k}} \sim \mathrm{Gamma}(2, 5) \tag{9}$$

$$\phi \sim \mathrm{Gamma}(0.1, 0.1) \tag{10}$$

Each model run consisted of four chains, each with 3,000 iterations and 1,500 warm-up runs; we used a thinning of every three samples and obtained a total number of 2,000 post-warmup samples. To assess whether the simulations converged, we checked the Gelman-Rubin potential scale reduction factor $\hat{R}$, which, if below 1.01, indicates that the Markov chains have converged (Kruschke, 2014). We also checked the effective number of independent samples $N_{\mathrm{eff}}$, indicating lower autocorrelation and higher efficiency of the convergence (McElreath, 2016).

### 2.2.2 Model selection

We trained the models using several different combinations of predictors to find the best balance between complexity and predictive accuracy. Our main motivation was to achieve a good balance of sufficiently detailed, but available data, which is often challenging (Meyer et al., 2013; Molinari et al., 2020). Each predictor in a multilevel model requires more than one parameter (i.e. $J$ group-level coefficients plus one hyperparameter). Hence, considering more parameters may offer small increases in predictive accuracy only at the risk of overfitting. We selected the model with the highest improvement compared to next simpler one, while retaining the same multi-level structure. On the one hand, testing all models possible without any underlying concept is far from good scientific practice and computationally inefficient; on the other hand, the predictors are rarely fully independent. Hence, we fitted candidate models in three steps of model comparison outlined below. We compare these models via the expected log pointwise predictive density (ELPD), which is the sum of a log-probability score of the predictive accuracy for unobserved data. The distribution of these unobserved data is unknown, but we can estimate the predictive accuracy with leave-one-out cross-validation (ELPD-LOO), which is the sum of the log-probability scores for the given data except for one data point at a time (Vehtari et al., 2017; McElreath, 2016). According to Vehtari (2020), an ELPD-LOO difference >4 may be relevant and should also be compared to the standard error of the difference. Hence, we selected models as follows:

1. We compared models with a gradually increasing number of predictors, based on the prior knowledge of predictor importance reported in a study using single-level linear regression by Mohor et al. (2020). This study considered water depth, for which data are the most widely available and adopted in flood loss models (Gerl et al., 2016), up to a maximum of twelve predictors (Table 1). For example, model 2 (named "fit2") has water depth (WD) and building area (BA) as predictors, while model 3 ("fit3") has the previous two plus contamination (Con) as predictors; model 12 ("fit12") has all twelve predictors (Table 1). The model candidate with an ELPD-LOO difference >4 compared to the previous candidate was selected for the next step.

2. For the model selected in step 1 – "fit_s1" with predictors $X^{(s1)} = \{x_1, \ldots, x_{s1}\}$, we compared models with $X^{(s1)}$ predictors plus one of the remaining predictors at a time, i.e., $\{X^{(s1)}\}, \{X^{(s1)}, x_{s1+1}\}, \{X^{(s1)}, x_{s1+2}\}, \ldots, \{X^{(s1)}, x_{12}\}$. All model candidates that present an ELPD-LOO difference larger than four and with a difference larger than its standard error were selected for step 3.

3. We compared the model candidates combining the selected candidates from step 2. If, for example, two different candidates $\{X^{(s1)}, x_{s1+a}\}$ and $\{X^{(s1)}, x_{s1+b}\}$ were selected, we compared the model candidates $\{X^{(s1)}\}, \{X^{(s1)}, x_{s1+a}\}$, $\{X^{(s1)}, x_{s1+b}\}, \{X^{(s1)}, x_{s1+a}, x_{s1+b}\}$. The model candidate with the least number of predictors and an ELPD-LOO difference >4 as well as a difference larger than the estimated standard error was selected eventually.

We compared all candidate models using leave-one-out cross-validation (LOO-CV) with Pareto smoothed importance sampling (PSIS-LOO), which is an out-of-sample estimator of predictive model accuracy (Vehtari et al., 2017), implemented in the R package `loo` (Vehtari et al., 2019).

Having identified the models with the most informative predictors, we checked for credible differences across levels using

the 95% highest density interval (HDI) of the marginal posterior distributions of the model parameters. We refer to regression intercepts and slopes as *credible* if their posterior HDIs exclude zero values, and to each pair of parameters as *credibly different* if 95% of the distribution of the difference of posterior estimates is above (or below) zero.

## 3   Results

We begin by reporting results form the model selection where we aimed at a compromise between model complexity, predictive accuracy, and data availability. For example, the generic model (Equation 5) has the lowest complexity with one ($K = 1$) predictor water depth (thus called "fit1"), and three group-levels for the regional model ($J = 3$). This model has eight parameters already, i.e. the population-level intercept ($\alpha_0$); three group-level intercepts ($\alpha_j$); three group-level coefficients for water depth ($\beta_{1,3}$); and parameter $\phi$. Candidate models with more predictors are more complex might fit the data better, but have a higher chance of missing input data at random. We test the increase in predictive capacity by adding predictors parsimoniously in light of this constraint.

### 3.1   Model selection

Judging from the predictive capacity using LOO-CV we arrived at a number of models worth further inspection. Table 3 shows how predictive accuracy in terms of the ELPD-LOO changes from the simplest water-depth model to eleven more complex candidates of the flood-type model (see Supplementary Material for other model variants). In this step, we consider a model to be significantly better if the difference of ELPD-LOO >4.

We find that models hardly improve beyond the complexity of model "fit6" (Table 3). Given that the choice of predictors may affect other predictors' contributions, we tested another set of models starting with the first six predictors but adding only one of the remaining predictors at a time, to evaluate if the order of adding predictors mattered (Table 4).

We find that "fit6+11" is the candidate model with the highest accuracy, though "fit6+7" is comparable (Table 4). We tested a final set of models with combinations of the best candidates, i.e. the predictors that showed significant increase among the further model candidates tested, namely predictors 6 (insured - Ins), 7 (perceived efficacy of PLPMs - Eff) and 11 (flood experience - Exp), added to the first five predictors (i.e. water depth, building area, contamination, duration, and Property-level Precautionary Measures (PLPM)). Note that fit5+6 equals fit6, but fit5+7 is not equal to fit7. The results for the Flood-type model are shown in Table 5 (for other model variants, see Figure 1 or Table S3).

Table 5 shows that two models are significantly better than "fit6" (fit5+6), i.e. "fit6+11" and "fit6+7+11". These two models are indistinguishable from each other in terms of their predictive accuracy, although model "fit6+11" has fewer predictors. We obtain similar results for other model variants (see Supplementary Material): for the regional model, "fit6+7" is also within the best candidates, while for the flood-event model adding more predictors hardly improves the predictive accuracy. In summary, we report that model "fit6+11" offered the best balance of complexity and performance among the model candidates considered.

**Table 3.** Comparison of flood-type model candidates of differing complexity and using their expected log pointwise predictive density (ELPD-LOO), ranked by increasing predictive accuracy, along with differences and their standard errors with reference to model "fit1" (see Table S1 for all model variants).

| Model | ELPD-LOO | ELPD-LOO difference | standard error of difference | Predictors |
|---|---|---|---|---|
| fit1 | 2018.7 | 0 | 0 | WD |
| fit2 | 2057.3 | 38.6 | 8.7 | WD+BA |
| fit3 | 2093.2 | 74.5 | 12.5 | WD+BA+Con |
| fit4 | 2098.1 | 79.4 | 12.8 | WD+BA+Con+Dur |
| fit5 | 2113.4 | 94.7 | 13.6 | WD+BA+Con+Dur+Pre |
| fit6 | 2124.0 | 105.3 | 14.1 | WD+BA+Con+Dur+Pre+Ins |
| fit8* | 2125.4 | 106.8 | 14.5 | WD+BA+Con+Dur+Pre+Ins+Eff+Eme |
| fit10* | 2125.9 | 107.2 | 14.8 | WD+BA+Con+Dur+Pre+Ins+Eff+Eme+Cel+Vel |
| fit9* | 2126.2 | 107.5 | 14.8 | WD+BA+Con+Dur+Pre+Ins+Eff+Eme+Cel |
| fit7* | 2127.0 | 108.3 | 14.5 | WD+BA+Con+Dur+Pre+Ins+Eff |
| fit11 | 2131.8 | 113.1 | 15.1 | WD+BA+Con+Dur+Pre+Ins+Eff+Eme+Cel+Vel+Exp |
| fit12* | 2134.3 | 115.6 | 15.3 | WD+BA+Con+Dur+Pre+Ins+Eff+Eme+Cel+Vel+Exp+BQ |

\* Difference between ELPD-LOO values between two subsequent models is <4

**Table 4.** Comparison of the flood-type model candidates by their difference in ELPD-LOO using the first six predictors plus one predictor at a time, ranked by increasing predictive accuracy, along with their differences and the standard error of the differences with reference to the model "fit6" (see Table S2 for all model variants)

| Model | ELPD-LOO | ELPD-LOO difference | standard error of difference | Predictors |
|---|---|---|---|---|
| fit6+8 | 2122.3 | -1.7 | 0.5 | WD+BA+Con+Dur+Pre+Ins+Eme |
| fit6+10 | 2123.2 | -0.9 | 1.4 | WD+BA+Con+Dur+Pre+Ins+Vel |
| fit6 | 2124.0 | 0 | 0 | WD+BA+Con+Dur+Pre+Ins |
| fit6+12 | 2124.2 | 0.2 | 2.0 | WD+BA+Con+Dur+Pre+Ins+BQ |
| fit6+9 | 2124.4 | 0.3 | 2.0 | WD+BA+Con+Dur+Pre+Ins+Cel |
| fit6+7 | 2127.0 | 3.0 | 3.5 | WD+BA+Con+Dur+Pre+Ins+Eff |
| fit6+11 * | 2130.8 | 6.7 | 3.9 | WD+BA+Con+Dur+Pre+Ins+Exp |

\* model with relevant improvement compared to others (elpd_diff > 4 and elpd_diff > se_diff)

## 3.2 Model diagnosis

We fit three multilevel models with the selected candidates (fit "6+11", i.e. water depth, building area, contamination, duration, PLPMs, insured, flood experience) in each of the flood-type, regional, and event model. All three multilevel models converged

**Table 5.** Comparison of Flood-type model candidates by their difference in ELPD-LOO using combinations of the first five predictors (fit5) plus predictors 6, 7, and 11, along with their differences and the standard error of the differences with reference to candidate model "fit5+6" (see Table S3 for all model variants)

| Model | ELPD-LOO difference | standard error of difference | ELPD-LOO | Predictors |
|---|---|---|---|---|
| fit5+7 | -6.2 | 6.1 | 2117.8 | WD+BA+Con+Dur+Pre+Eff |
| fit5+11 * | -3.5 | 6.4 | 2120.5 | WD+BA+Con+Dur+Pre+Exp |
| fit6 * | 0 | 0 | 2124.0 | WD+BA+Con+Dur+Pre+Ins |
| fit5+7+11 * | 0.1 | 7.4 | 2124.1 | WD+BA+Con+Dur+Pre+Eff+Exp |
| fit6+7 * | 3.0 | 3.5 | 2127.0 | WD+BA+Con+Dur+Pre+Ins+Eff |
| fit6+11 | 6.7 | 3.9 | 2130.8 | WD+BA+Con+Dur+Pre+Ins+Exp |
| fit6+7+11 | 9.6 | 5.4 | 2133.6 | WD+BA+Con+Dur+Pre+Ins+Eff+Exp |

*models with predictive accuracy that is indistinguishable from that of the reference model fit6

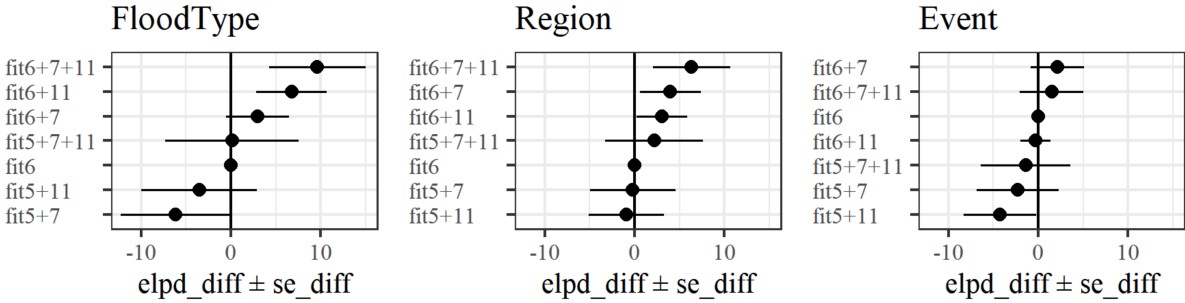

**Figure 1.** Comparison of model candidates by their difference in ELPD-LOO using combinations of the first five predictors (fit5) plus predictors 6, 7, and 11, along with their differences and the standard error of the differences with reference to candidate model "fit6" for each model variant

($\hat{R} < 1.004$) with effective sample sizes $N_{\text{eff}}$ from 1,164 to 1,273 (out of 2,000 samples). The multilevel model was trained with 70% of the dataset that was drawn through random sampling maintaining the proportion of group levels, totalling 1,269 data points without missing data. The remaining 30% of the data were used for a performance check (Table 6).

We also ran posterior predictive checks by comparing the observed distribution of the loss ratio with the posterior predictive distribution drawn from the training and the test data (Figure 2). The shapes of the posterior predictive distributions align well with the observed data, indicating that the models suitably simulate the response variable.

### 3.3 The roles of flood type, affected region, and flood event

In this section we show the group-level coefficient estimate intervals of each model and whether they are credibly different for different groups. We report the highest density interval (HDI) of the posterior model weights and compare these estimates

**Table 6.** Performance indicators over mean values of the posterior predictive distribution (median of performance indicators over the full posterior predictive distribution) and convergence indicators of the three model variants. RMSE = root mean squared error; MAE = median absolute error; $\hat{R}$ = Gelman-Rubin potential scale reduction factor; $N_{\text{eff}}$ = effective sample size.

| Model | Dataset | RMSE | MAE | highest $\hat{R}$ | lowest $N_{\text{eff}}$ |
|---|---|---|---|---|---|
| Flood-type model | Train | 0.102 (0.138) | 0.046 (0.053) | 1.003 | 1,236 |
| | Test | 0.108 (0.143) | 0.044 (0.055) | | |
| Regional model | Train | 0.104 (0.140) | 0.045 (0.054) | 1.004 | 1,273 |
| | Test | 0.110 (0.145) | 0.045 (0.056) | | |
| Event model | Train | 0.103 (0.139) | 0.045 (0.053) | 1.004 | 1,164 |
| | Test | 0.111 (0.144) | 0.043 (0.055) | | |

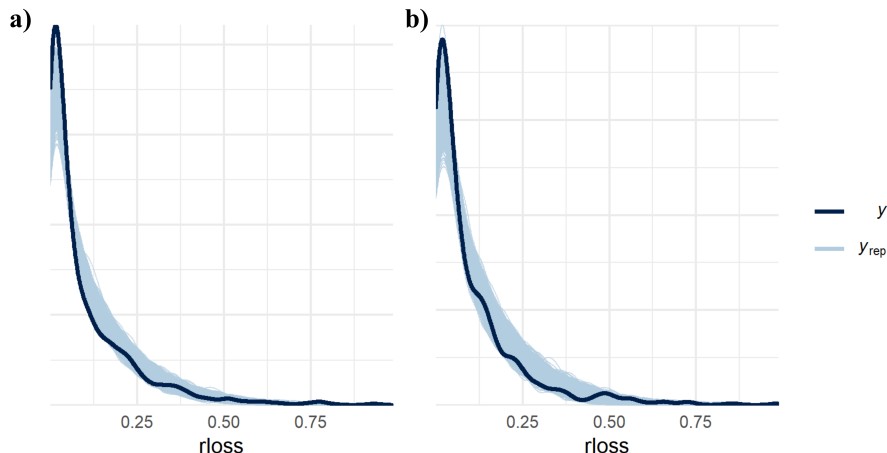

**Figure 2.** Density plot of observed loss ratio ($y$) and simulations drawn from posterior predictive distribution ($y_{\text{rep}}$) over a) training ($n = 1{,}269$) and b) testing ($n = 543$) data with flood-type model

between the groups of each model. The models use a inverse-logit transformation over the linear regression (Equation 5) to transform any real value to the unit interval. For example, a population-level intercept $\alpha_0 = -2.37$ means that, holding all predictors fixed at zero (or their average), $\text{logit}^{-1}(-2.37 + 0) = 0.085$; hence the estimated average loss ratio is 8.5%. Positive (negative) coefficient estimates of each predictor will result in a larger (smaller) loss ratio from the average on the log-odds scale.

### 3.3.1 Flood-type model

Figure 3 shows the 95% HDI of the predictor weights grouped by flood types (flood-type model) compared to that of the pooled model. The groups of surface water and groundwater flooding have fewer data (levee breaches, $n = 258$; riverine $n =$

683; surface water $n = 152$; groundwater $n = 176$) and thus more uncertain parameter estimates with wider HDIs (Figure 3), although several of these estimates are credible. Six out of seven predictors, i.e. water depth, contamination, duration, PLPMs, insured, and flood experience, have at least one pair of flood types with credibly different estimates. In these cases the 95% HDI of the differences between the posterior estimates is above or below zero. Most estimates are credibly positive or negative, and only a few estimates HDI 95% contain zero.

For example, the standardised group-level intercepts $(\alpha_0 + \alpha_j)$ that estimate the loss ratio for average predictor values, are credibly smaller for groundwater floods than for other flood types. Water depth has a credibly higher weight for levee breaches, i.e. the effect of each unit increase in water depth on the loss ratio is higher for levee breaches, than for surface water floods (Figure 3-b, Table 7). In most cases, the differences show a higher effect of levee breaches over other flood types. The contamination effect of surface water floods is also credibly higher than of riverine floods, and the effect of riverine flood duration

credibly outweighs that of groundwater-flood duration.

The effects of flood duration (Figure 3-e), the insurance indicator (Figure 3-g), and the flood-experience indicator (Figure 3-h) remain inconclusive concerning surface water or groundwater floods. Similarly, flood PLPM implementation (Figure 3-f) is an ambiguous predictor of relative loss caused by levee breach or groundwater floods.

### 3.3.2 Regional model

Figure 4 shows the HDI 95% of the regression coefficients if we group the loss data across various regions of Germany. The group of flood-affected households from western and northern Germany is the smallest (south $n = 337$; east $n = 865$; west and north $n = 67$), so the posterior parameter estimates are less certain and, in most cases, inconclusive for this part of the country. Similar to the flood-type model, all estimates are credibly different from zero for water depth (Figure 4-b). The HDIs of all predictors overlap, i.e. there are hardly credible difference across regions under this model. The only estimate that is ambiguous

in the southern region is that for flood experience (Figure 4-h).

### 3.3.3 Event model

Figure 5 shows the HDI 95% of the posterior regression weights if grouping the data across individual flood events indexed by years. The data subsets of flood-affected households in 2002 and 2013 are largest, (2002 = 571 cases; 2005 = 56; 2006 = 30; 2010 = 141; 2011 = 66; 2013 = 405), hence their estimates are more certain than those for other events. Similar to the results of

280 the regional grouping, we notice a large overlap of parameter estimates across individual floods without credible differences. Estimates of the intercept (Figure 5-a) are highest for 2002 and 2013, whereas the other, lower estimates overlap, except for 2010 and 2011 that are also distinct from each other (Table 7). This result underlines that the floods of 2002 and 2013 were more damaging than other events on average.

The estimates of water depth (Figure 5-b), where the 95% HDIs for 2002, 2010, and 2013 are credibly higher than for 2005.

The HDI for 2013 is also credibly higher than that for 2011, while other pairs of estimates overlap (Table 7). The coefficient estimates for duration and the PLPMs implementation (Figure 5-e and -f) for 2002 surpass the estimates for 2010, which in turn are ambiguous. The estimate for the insurance indicator of 2013 exceeds that for 2005, although all 95% HDIs except for

**Table 7.** Credibly different pairs of estimates with 95% probability

| Comparison | Predictor | Median of differences | % above 0 |
|---|---|---|---|
| Levee Breach-Groundwater | Intercept | 0.323 | 99.4% |
| Riverine-Groundwater | Intercept | 0.212 | 98.6% |
| Surface-Groundwater | Intercept | 0.210 | 96.7% |
| Levee Breach-Surface | Water Depth | 0.155 | 98.4% |
| Riverine-Surface | Contamination | -0.167 | 1.6% |
| Riverine-Groundwater | Duration | 0.114 | 95.2% |
| Levee Breach-Riverine | PLPMs implementation | 0.162 | 99.0% |
| Levee Breach-Surface | PLPMs implementation | 0.207 | 98.6% |
| Levee Breach-Riverine | Insured | 0.107 | 96.7% |
| Levee Breach-Surface | Insured | 0.213 | 99.6% |
| Levee Breach-Groundwater | Insured | 0.186 | 98.9% |
| Levee Breach-Surface | Flood Experience | -0.228 | 0.6% |
| Levee Breach-Groundwater | Flood Experience | -0.195 | 1.9% |
| 2002-2005 | Intercept | 0.521 | 100.0% |
| 2002-2006 | Intercept | 0.448 | 98.7% |
| 2002-2010 | Intercept | 0.261 | 99.6% |
| 2002-2011 | Intercept | 0.612 | 99.9% |
| 2005-2013 | Intercept | -0.517 | 0.2% |
| 2006-2013 | Intercept | -0.447 | 1.2% |
| 2010-2011 | Intercept | 0.346 | 95.2% * |
| 2010-2013 | Intercept | -0.259 | 0.7% |
| 2011-2013 | Intercept | -0.609 | 0.1% |
| 2002-2005 | Water Depth | 0.343 | 99.5% |
| 2005-2010 | Water Depth | -0.369 | 0.7% |
| 2005-2013 | Water Depth | -0.394 | 0.2% |
| 2011-2013 | Water Depth | -0.259 | 3.1% * |
| 2002-2010 | Duration | 0.175 | 98.7% |
| 2002-2010 | PLPMs implementation | -0.179 | 1.8% |
| 2005-2013 | Insured | -0.157 | 4.5% * |

* Although the one-sided hypothesis is satisfied, with 95% of the posterior distribution being above, or below, zero, the HDI 95% of the distribution of the differences contains zero

the one for 2013 contain zero. We note that many parameter estimates cover mostly small values; especially flood experience (Figure 5-h) is an inconclusive predictor in contrast to the other models (Flood-type model or Regional model) that showed

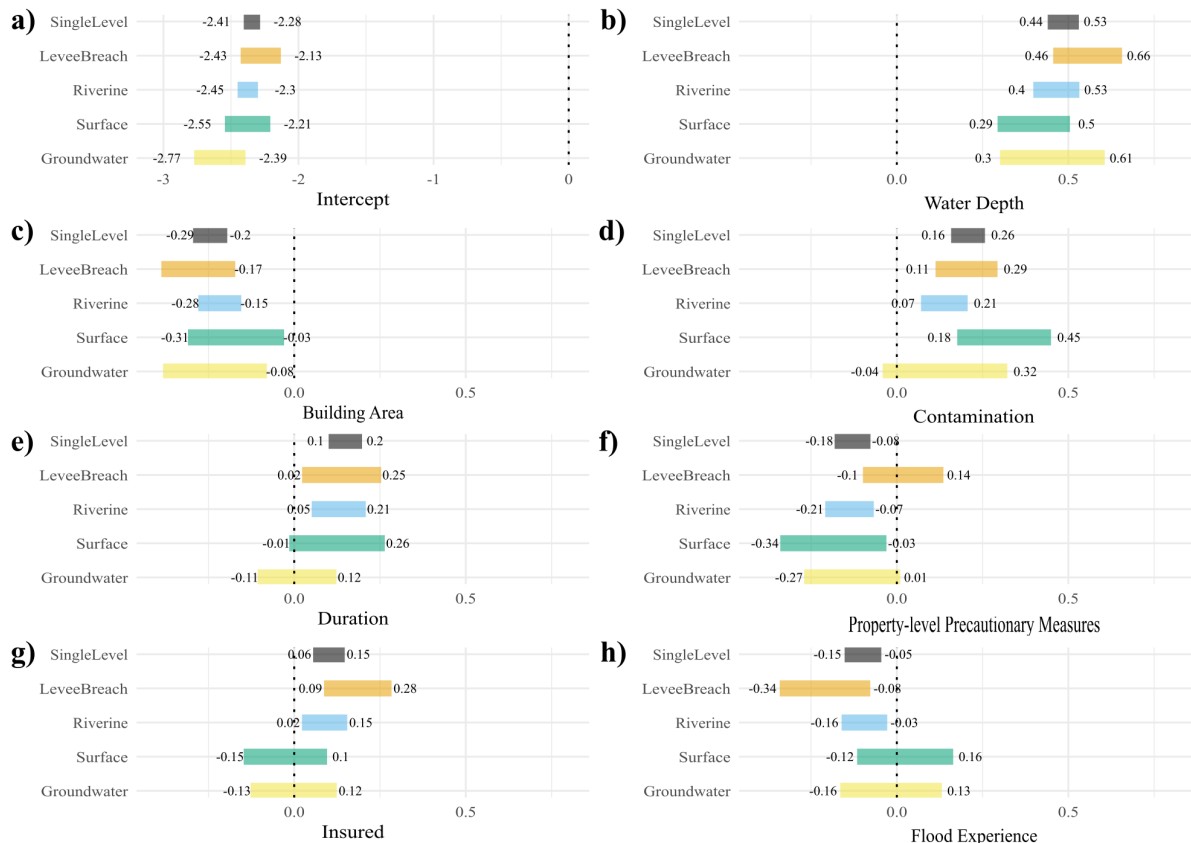

**Figure 3.** HDI 95% of regression estimates of the Flood-type model (across four flood types, coloured segments) and the single-level model (black segments). The intercept is the sum of the population-level effect (common across levels) and group-level effects (for each flood type)

credible estimates for at least one group. There is no clear tendency of estimates increasing or decreasing with time; on the contrary, there is a large overlap across most events and predictors.

## 4 Discussion

We trained three variants of a Bayesian multilevel model to test whether flood type, regions within Germany, or flood events make a case for differing predictor influences on flood loss concerning these groups. The models help us to identify the factors most relevant for flood loss estimation and to assess whether there are credible differences between these contributions to the estimated loss ratio. In other words, the models show how considering these groups is a useful step towards improved model transferability.

After comparing the predictive accuracy estimates of models with different sets of predictors, we selected the model "fit 6+11" that uses water depth, building area, contamination, duration, PLPMs, insurance, and previous flood experience as predictors.

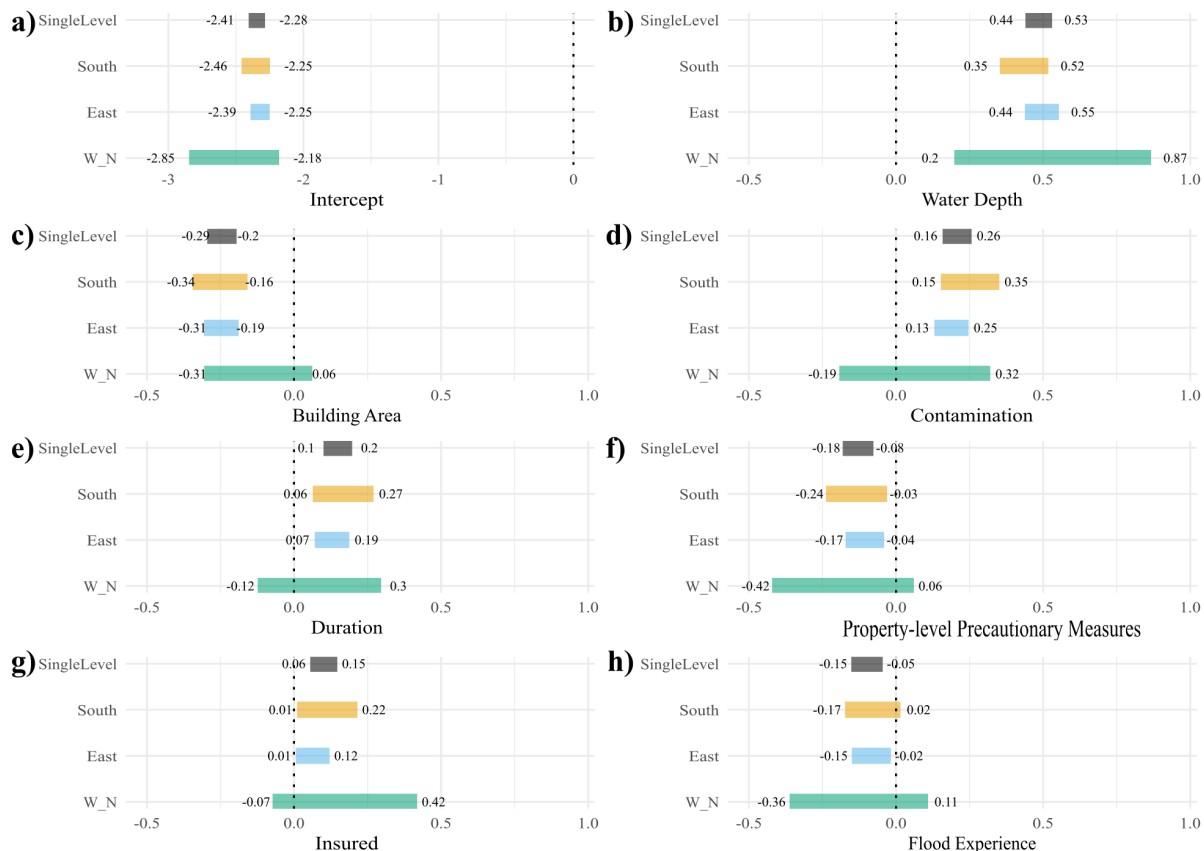

**Figure 4.** HDI 95% of regression estimates of the Regional model (across three regions, coloured segments) and the single-level model (black segments). The $Intercept$ is the sum of the population-level effect (common across levels) and group-level effects (for each region)

Considering that we aim to explore the role of predictors in estimating flood losses, rather than finding the best fit model, chains convergence and posterior predictive checks are a necessary step before interpreting the fitted model (Gabry et al., 2019; Gelman et al., 2020). The three model variants trained with 1,269 datapoints, and sampled with four chains each, converged well, with Gelman-Rubin scales below 1.004 (ideal values are <1.01) and effective sample size ratios above 0.58 (ideal values are >0.5). Visual assessment of the predictive posterior density plot is an important step, whether the model

generates data similar to the observed data. Figure 2 shows that the model replicates well the data distribution, and visual inspection confirmed only unimodal estimates.

Our results show that, for most cases across regions or across flood events, the posterior regression weights are hardly different. Therefore, distinguishing groups, at least in the form here implemented, adds little information over a pooled model taking into account all of the data. Out of the training dataset of 1,269 data points, the groups contained much smaller (<200 to <50)

samples, thus giving rise to higher uncertainties regardless of the shrinkage of coefficient estimates in a Bayesian multilevel model towards the pooled means. Credible differences across estimates are found mostly if considering flood types and this

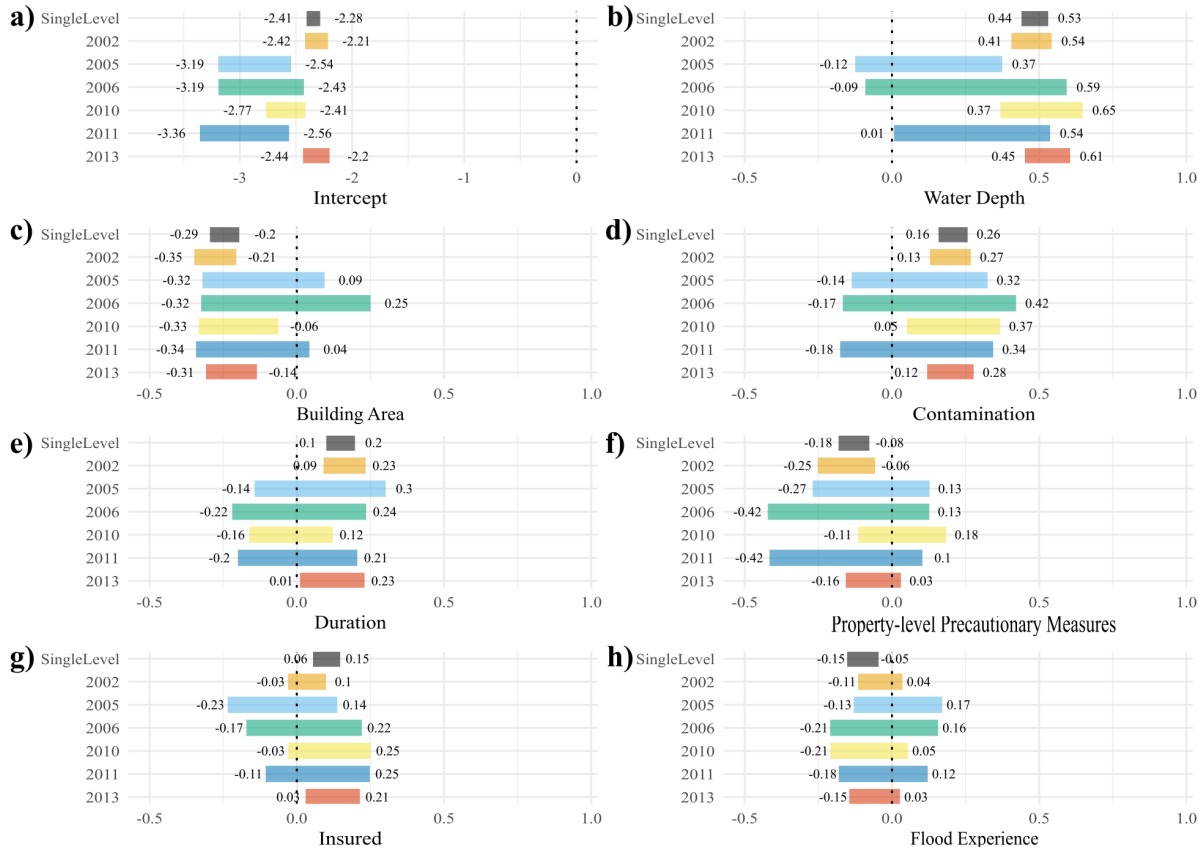

**Figure 5.** HDI 95% of regression estimates of the flood-event model (each event coded by colour) and the single-level model (black bars). The intercept is the sum of the population-level effect (common across levels) and group-level effects (for each event)

grouping also involves more balanced subsets. The estimated coefficients for loss-ratio modelling across flood events and regions are mostly inconclusive. However, especially in western and northern Germany, the 2005, the 2006, or the 2011 flood events return many inconclusive parameter weights, likely owing to the much fewer data points. Leaving these very uncertain estimates aside, we can observe several instructive patterns.

We note that the higher the water depth, the contamination of the floodwater, or the duration a building is inundated, the higher is the loss ratio, assuming all other predictors fixed. This is a simple expectation (Kellermann et al., 2020) being confirmed, also showing that these predictors add information to the model (see Figures 3, 4, 5-b and -e). Next, the larger the building, the lower the relative damage. This is also reasonable, since larger buildings, which mostly have more floors, would experience lower relative damage with all else kept constant (Thieken et al., 2005). We also find that the more recently a household experienced a flood, the lower the relative damage. People who experienced more recent floods (scored higher in the flood-experience indicator), on average, appear to be better acquainted with how to act before and during a flood, thus reducing its risks and direct impacts. The indicator of whether the household had an insurance has mostly positive weights, although

often also ones that are ambiguous. This result is in agreement with previous studies showing an unclear effect of insurance coverage on loss reduction (Surminski and Thieken, 2017). Finally, the indicator of PLPM implementation also has a mostly negative weight on predicting the loss ratio. This may mean that the more PLPMs implemented, the lower the relative damage, as shown by Kreibich et al. (2005) and Hudson et al. (2014). However, this indicator encompasses several measures so that the damage reducing effect of each such measure in different flood situations is intractable. Hence, this result only shows a general tendency that PLPMs reduce relative damage, but to a much-varied degree that deserves further research.

Although previous work has indicated a more intense flood events in eastern than in southern Germany, except for the 2005 flood (Schröter et al., 2015), we found no credibly different estimates in our regional model (Figure 4). It is likely that different precaution strategy of residents matter here, as more people in the East have relied on insurance (Thieken, 2018), although the effect of having insurance on flood losses remains unclear; the effect of PLPMs also overlaps across estimates for southern and eastern Germany.

Despite the large overlap across estimates of the flood-event model, we find that the estimates for 2002, 2010, and 2013 for water depth and contamination are larger and more credible, reflecting also larger average losses reported by the households (Table S4). Although the 2006 subsample had a large average flood duration (Table S4), it still returns a highly uncertain coefficient estimate. The severe Central European flood of August 2002 in Germany mainly affected the rivers Danube and Elbe, and only a few households had implemented PLPMs or had previous flood experience (Thieken et al., 2007); this situation changed for later floods (Kienzler et al., 2015). Consequently, the implemented PLPMs made a larger difference for the flood of 2002 (the only credible estimate), whilst the role of previous flood experience remains ambiguous in the models. In contrast, as insurance coverage increased over time, only the 2013 estimate was credibly positive; having an insurance seems to be linked to a higher loss ratio. This finding that insurance has a positive effect—though only for the later event—may indicate that either moral hazard has increased (i.e. insured people declare more damage) or that more people in risk-prone areas have purchased insurance coverage against flooding. The latter would indicate that risk communication was partly successful. To confirm this, however, not only would the increase in insurance uptake need to be checked, but it would also need to be crossed with flood risk zones. This is a task for future work.

We emphasise that each event and each region of Germany contained mixed flood types (or pathways). For most predictors, the factors' effects are much clearer across flood types. This reinforces the notion that their importance varies across flood types. Given that mixed flood types were reported in all regions and years in our dataset, this might be the reason the predictors effects are also less certain and overlapping across regions and years.

It is plausible that the effects of some variables are influenced by others, whether included or ignored in our initial set. Only a few studies have so far directly compared the effect of predictors of flood loss ratio across groups in the data, such as flood types, events, or places. Two of them, i.e. Vogel et al. (2018) and Sairam et al. (2019), used a similar dataset. Although these studies adopted different model structures, we compare below our results.

Sairam et al. (2019) trained and compared hierarchical Bayesian models for flood loss estimation as we did here, but they considered only water depth as a single predictor. Sairam et al. (2019) tested as grouping variables the river basins, the event years, and a combination of both, and concluded that the latter had the best predictive accuracy. This approach, however, masks

the weight of effects across areas or events, as both effects are bundled. Despite the differences in the grouping, similarly, Sairam et al. (2019) found significant differences between regression slopes, but not across intercepts, reinforcing that using flood type as grouping variable seems to be more relevant compared to flood event or region.

Vogel et al. (2018) trained Markov Blankets (MBs) for estimating the flood loss ratio for different flood types and different events, separately. MBs are the smallest components of Bayesian Networks (BNs) and contain all variables that are relevant, out of the originally chosen, for predicting the targeted variable (Vogel et al., 2018). Therefore, we cannot compare estimates, but only the predictors set selection. We selected the predictors across all levels, which makes a direct comparison difficult, trained independently. Still, we observe some similarities between ours and the results by Vogel et al. (2018). For example, Vogel et al. (2018) showed that previous flood experience and flood duration are both relevant for households affected by levee breaches, whereas building size, which is correlated to building area, is relevant for riverine floods. For the MBs trained for each flood event, Vogel et al. (2018) found water depth to be a common predictor for all events, except for the flood of 2011, which comprises one of the smallest subsamples, in which previous flood experience was the only predictor selected, in contrast to our findings. Our very uncertain estimates across event years for this predictor suggests it may be biased and deserve more attention before dismissing all estimates with HDI containing zero. More data should be collected or predictors could represented differently, for example as a monotonic effect.

Data availability, especially regarding preparedness indicators, is a possible limitation to transferring flood loss models and their use for ex-ante loss estimation. While these indicators have been deemed relevant for loss prediction, they are rarely collected and are often unavailable in a suitable form. An alternative is to use proxy data, for example the aggregated insurance coverage for Germany monitored by the German Insurance Association (GDV, 2018) as proxy for household insurance, a good flood event database could be a rough estimate of flood experience for a specific region, or the precautionary behaviour of flood-affected residents (Bubeck et al., 2020) could be used as a prior estimate of PLPMs implementation. Nonetheless, the role of data availability is directly captured in our models in terms of (un-)certainty of posterior parameter estimates. Bayesian models excel in situations where data are limited, but also express the associated uncertainties.

When addressing transferability, we seek models that can generalize well and go beyond local or case-specific data. Wagenaar et al. (2018) trained two flood loss models using data from two different countries (Germany and the Netherlands) and tested how well each model could predict losses in the other country. They found that the number of flood events in the data was more important than simply the number of reported flood loss cases. Although we trained our models with data from a single country, the data used by Wagenaar et al. (2018) for Germany, comprises six event years across twelve federal states, four river basins (Danube, Rhine, Elbe, and Weser) and four flood types. We expanded on this approach by training models on data from different flood-event years, different flood types, and different regions, thus allowing for a broad range of environmental, administrative, and socio-economic conditions (representing at least Central Europe) that we treat explicitly as grouping levels in our analysis. We argue that exploring these model variants provides more clarity about whether we should use simple average models or more specific multi-level models to be able to transfer predicted loss estimates to new regions, flood types or other structures in the data.

## 5 Conclusions

Previous studies have indicated that the major damaging processes during floods may differ by flood type, event, and affected region. To better understand these differences and improve the transferability of flood loss models, we trained and tested Bayesian multilevel models for estimating relative flood losses to residential buildings.

Our model selection identified seven predictors addressing the flood magnitude (water depth, contamination, and duration), the building size (building area), and preparedness of the household (previous experience, insurance, and an indicator of implemented PLPMs). For at least one group, all predictors show credible posterior estimates HDI 90%. This result confirms that all these predictors can aid flood loss ratio estimation, and reinforces the need to collect data after new flood events. This repeated updating is at the core of Bayesian models, which can also handle missing data, account for uncertainty intrinsically, and are effectively finding a compromise between existing models and new data. We argue that this strategy might pave one way for transferring flood loss models more widely.

Credibly different estimates were found for six out of seven predictors across flood type, region, and event year, namely: water depth, contamination, duration, implementation of property-level precautionary measures, insurance, and previous flood experience. The Bayesian multilevel model grouped by flood type is the most informative of these three model variants, featuring the most pronounced differences in the contributions of each predictor. Despite credible differences between different flood events, the large uncertainties in the posterior estimates of the regional and the event models likely indicate that several flood types may have mixed during a single flood event or region, thus making it difficult to disentangle individual controls better. In any case, the dataset hardly caters to reveal fully the underlying physical controls on flood losses.

Our results encourage using pooled data on flood events and regions, and thus mark some transferability in this regard, judging from the minute differences in the posterior regression weights. The data indicate, however, that flood loss modelling should consider different flood types explicitly. We acknowledge that other groups in the data or a different set of predictors could improve predictions further, but recommend strategies that make use of previous knowledge as much as possible. We conclude by reporting that grouping models by flood type adds information and transferability to flood loss estimation and motivate more research into this direction.

*Data availability.* The survey data are owned by the second author. Data from the 2002 event can be provided only upon request. Data from the 2005, 2006, 2010, 2011, and 2013 events are available via the German flood loss database HOWAS21 (http://howas21.gfz-potsdam.de/howas21/).

*Author contributions.* Conceptualisation of the study: all authors; Model development: Guilherme S. Mohor and Oliver Korup; Code development: Guilherme S. Mohor; Result analysis: all authors; Writing: all authors.

*Competing interests.* The authors allege no competing interests.

*Acknowledgements.* This work received financial support from the DAAD (Graduate School Scholarship Programme, 2017 - ID 57320205). Besides original resources from the partners, additional funds for data collection were provided by the German Ministry for Education and Research (BMBF) in the framework of the following research projects: DFNK 01SFR9969/5, MEDIS 0330688, and Flood 2013 13N13017. The surveys were conducted by a joint venture between the GeoForschungsZentrum Potsdam, the Deutsche Rückversicherung AG, Düsseldorf, and the University of Potsdam. We thank Meike Müller, Ina Pech, Sarah Kienzler, and Heidi Kreibich for their contributions to the survey design and data processing.

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
