# Peer review of "Residential flood loss estimated from Bayesian multilevel models"

_Natural Hazards and Earth System Sciences, 2020_

## Referee Comment (RC1) · Nivedita Sairam (Referee) · 17 Jan 2021

The paper presents a Bayesian multi-level model to estimate residential flood losses. The study is meticulously undertaken and the manuscript is very clear. The authors infer that flood source/type is an important aspect that causes variability in damage model parameters. This is determined based on significant differences in damage model parameters/hyper-parameters across flood types such as levee breach, riverine, surface and groundwater flooding. Additionally, the study claims that the model with predictors - water depth, building area, contamination, duration, precaution, insurance uptake and flood experience performs the best.

Please address the following comments:

[Figure]

Abstract: 1. Line 11: 'may complicate' lacks clarity. Please rephrase to directly mention the impact of the developed modelling approach

Introduction: 2. Line 31: Abbreviation - BMM is not used anywhere in the paper 3. Lines 36-45: This paragraph about data could be moved to the section 2.1 – Data

Data and Methods: 4. In table 2, Please also provide the split of data samples across events and regions for each flood type. The different flood types may be relevant event characteristics when split across regions. 5. In the model selection, please explicitly explain what 'little gain' means? From the values, I understand that elpd diff > 4 when adding each variable is considered as significant improvement. Is it correct? (https://avehtari.github.io/modelselection/CV-FAQ.html#15_How_to_interpret_in_Standard_error_(SE)_of_elpd_difference_(elpd_diff)) Also, please refer to STAN/BRMS forums for considerations while choosing models based on LOO-ELPD differences. Studies commonly consider elpd differences above 2SE as a significant improvement (e.g. https://iopscience.iop.org/article/10.1088/1748-9326/ab4937). Here, the STAN developers recommend 4SE as a safe threshold (https://discourse.mc-stan.org/t/loo-comparison-in-reference-to-standard-error/4009/2). However, since it is a measure of balance between bias and variance, I would like to leave it to the authors' discretion. 6. Please provide (in main manuscript) the ELPD differences and SE across the four model types – single level, flood type, event and region for fit6+11 (From SI, table S3). Are there significant differences? Which of the models perform the best?

Discussion: 7. Please present some discussion on the model diagnosis (section 3.2) 8. The concept of transferability is only discussed in lines 260-263. Please provide further analysis or information regarding how the model addresses the transferability challenge.

SI: 9. Table S3 – rephrase Year to Event 10. Please mention the corresponding SI tables in the respective sections (main manuscript). Where are tables S4 and S5 relevant?
* * *

---

## Referee Comment (RC2) · Anonymous Referee #2 · 24 Jan 2021

This paper describes the development of a Bayesian multilevel model for flood damage estimation. These two step models first group observations by event, flood type or region and then build separate models. The study showed that grouping by flood type is most useful for developing transferable flood damage models. The study seems to be carried out well and the writing is generally good. I therefore just have some minor suggestions for improvements:

• One of the main conclusions seems to be that when developing transferable flood damage models it works best to select models by flood type rather than by event or region. This observation is very interesting but this is based on a dataset of just German data. I can imagine that in a more international setting the regional difference might become more important than the flood type differences. I think this needs to be

emphasized in the conclusions and I think the paper should therefore more promote the method than the finding (which I expect to be specific to this dataset of a relatively homogenous region).

âǎć Can you maybe explain better why you go for a multilevel approach rather than just adding variables like flood type, region and event to the dataset? You can then use variable importance to see how much these variables add. In other words can you clarify the added value of this approach better compared to this obvious/simpler alternative approach?

âǎć In the first sentence of the abstract you note that preparedness is typically ignored. I agree with this statement but its not really what this paper is about and by adding it to the first sentence of the abstract you confuse the reader. So I advice moving this statement.

âǎć Maybe also mention synthetic models in the introduction.

âǎć Line 26: The introduction frames that having a lot of detailed information automatically leads to overfitting and reasons that you therefore need multi-level models. This is not necessarily true, overfitting can be controlled in almost all data-driven methods so its possible to produce more general models with detailed data. Multi-level models are just another way of doing this not the only way.

âǎć The explanation in 2.2.1 and 2.2.2 is a bit difficult to follow. Could you try improve the explanation, maybe using a figure.

âǎć I think the title of 2.2.2 should be more like model tuning rather than model comparison, because you really use the same model but with different settings.

---

## Author Comment (AC1) · 26 Feb 2021

We would like to thank the referees for carefully reading our manuscript and for their constructive comments and suggestions. Please find below our point-by-point responses. If a change in the manuscript will be made, we explicitly say so and present the new excerpts below in red.

**Reviewer 1:**

"The paper presents a Bayesian multi-level model to estimate residential flood losses. The study is meticulously undertaken and the manuscript is very clear. The authors infer that flood source/type is an important aspect that causes variability in damage model parameters. This is determined based on significant differences in damage model parameters/hyper-parameters across flood types such as levee breach, riverine, surface and groundwater flooding. Additionally, the study claims that the model with predictors - water depth, building area, contamination, duration, precaution, insurance uptake and flood experience performs the best."

We thank Dr. Sairam for this positive and encouraging assessment of our study. Please find below the answers to each comment.

***R1-C1. Line 11: 'may complicate' lacks clarity. Please rephrase to directly mention the impact of the developed modelling approach.***

A: We rephrased this to:
"We argue that failing to do so may unduly generalize the model and systematically bias loss estimations from empirical data."

***R1-C2. Line 31: Abbreviation - BMM is not used anywhere in the paper***

A: The abbreviation was used in previous versions of the manuscript. We apologize for the confusion and removed the abbreviation.

***R1-C3. Lines 36-45: This paragraph about data could be moved to the section 2.1 – Data***

A: We agree with the suggestion, and moved the original paragraph to section 2.1., replacing it with a simpler introduction to the dataset. The paragraph now reads:

"In this study, we use survey data from households affected by large floods throughout Germany between 2002 and 2013 (Thieken et al., 2017). These data go beyond addressing physical inundation characteristics by offering a broad view of the damaging process including the flood types that affected the households (i.e., floods from levee breaches, riverine floods, surface water floods, or rising groundwater floods). Mohor et al. (2020) used this database to explore ... "

***R1-C4. In table 2, Please also provide the split of data samples across events and regions for each flood type. The different flood types may be relevant event characteristics when split across regions.***

A: We split the data guided by the intersect of flood type, event year and region, thus conserving the proportions across each group: the training set contains ~70% of each subgroup in Table 2 (original numbers). Thus, we found it best to replace the numbers in Table 2 by the size of the training set rather than showing both training and test datasets. The revised Table 2 looks like this:

Table 2. Number of instances in the training set used across grouping variables of flood type, region, and event year (n = 1269)

| Event\ Flood Types | Levee Breach | Riverine | Surface | Groundwater | n |
|---|---|---|---|---|---|
| 2002 | 110 | 252 | 103 | 106 | 571 |
| 2005 | 8 | 35 | 7 | 6 | 56 |
| 2006 | 0 | 25 | 2 | 3 | 30 |
| 2010 | 31 | 86 | 19 | 5 | 141 |
| 2011 | 1 | 49 | 5 | 11 | 66 |
| 2013 | 108 | 236 | 16 | 45 | 405 |
| Regions of Germany | | | | | |
| South | 52 | 174 | 53 | 58 | 337 |
| East | 205 | 469 | 80 | 111 | 865 |
| West and North (W+N) | 1 | 40 | 19 | 7 | 67 |
| Sum (n) | 258 | 683 | 152 | 176 | 1269 |

**R1-C5. In the model selection, please explicitly explain what 'little gain' means? From the values, I understand that elpd diff > 4 when adding each variable is considered as significant improvement. Is it correct? (https://avehtari.github.io/modelselection/CVFAQ.html#15_How_to_interpret_in_Standard_error_(SE)_of_elpd_difference_(elpd_diff)) Also, please refer to STAN/BRMS forums for considerations while choosing models based on LOO-ELPD differences. Studies commonly consider elpd differences above 2SE as a significant improvement (e.g. https://iopscience.iop.org/article/10.1088/1748-9326/ab4937). Here, the STAN developers recommend 4SE as a safe threshold (https://discourse.mc-stan.org/t/loo-comparison-in-reference-to-standarderror/ 4009/2). However, since it is a measure of balance between bias and variance, I would like to leave it to the authors' discretion.**

A: Thank you for this helpful comment and the further information. To avoid an automatic forward selection of the most suitable model, we followed three steps of comparison that we presented in Tables 3 + SI1, 4 + SI2 and 5 + SI3. In general, we avoided being too strict in the selection because we wanted to keep the same set of predictors for all model variants, i.e., the variants in which we considered different grouping variables. One advantage of Bayesian inference is the explicit treatment of uncertainty; our interpretations are based on full distributions rather than point estimates. Thus, we focused on exploring the differences across posterior parameter values rather than simply seeking the "best" model.

We note that Vehtari's CV-FAQ (https://avehtari.github.io/modelselection/CVFAQ.html) considers an elpd_diff > 4 to be relevant. Our model selection consisted of three steps following this consideration as a guideline, although this was perhaps less clear in our original manuscript. We made this point clearer by restructuring the Tables and extending the explanation of the three steps. In the first step, only the elpd_diff > 4 was considered. In the second and third step, we added the relationship between elpd_diff and its SE to the criteria. Assuming the elpd_diff is Gaussian

indicates that using two standard errors can be a meaningful measure of potential overlap. In the discussions of the STAN developers and users, again, these measures of model comparison are already understood as containing some variance. We were less strict in our study for the reasons outlined above. In the manuscript, we thus improved the tables by changing the reference model and extended the description of the three steps explicating the used criteria. The following paragraphs replaces lines 156-169:

"On the one hand, testing all models possible without any underlying concept is far from good scientific practice and computationally inefficient; on the other hand, predictors are rarely fully independent. Hence, we fitted candidate models in three steps of model comparison outlined below. We compare the model candidates in each step via the expected log pointwise predictive density (ELPD), which is the sum of a log-probability score of the predictive accuracy for unobserved data. The distribution of these unobserved data is unknown, but we can estimate the predictive accuracy with leave-one-out cross-validation (ELPD-LOO), which is the sum of the log-probability scores for the given data except for one data point at a time (Vehtari et al., 2017; McElreath, 2016). According to Vehtari (2020), an ELPD-LOO difference >4 may be relevant and should also be compared to the standard error of the difference. Hence, we selected models as follows:

1- We compared models with a gradually increasing number of predictors, based on the prior knowledge of predictor importance reported in a study using single-level linear regression by Mohor et al. (2020). This study considered water depth, for which data are the most widely available and adopted in flood loss models (Gerl et al., 2016) up to a maximum of twelve predictors (Table 1). For example, model 2 (named "fit2") has water depth (WD) and building area (BA) as predictors, while model 3 ("fit3") has the previous two plus contamination (Con) as predictors; model 12 ("fit12") has all twelve predictors (Table 1). The model candidate with an ELPD-LOO difference >4 compared to the previous candidate was selected for the next step.

2 – For the model selected in step 1 – "fit_s1" with predictors $X^{(s1)} = \{x_1, \ldots, x_{s1}\}$, we compared models with $X^{(s1)}$ predictors plus one of the remaining predictors at a time, i.e., $\{X^{(s1)}\}$, $\{X^{(s1)}, x_{s1+1}\}$, $\{X^{(s1)}, x_{s1+2}\}$, $\ldots$, $\{X^{(s1)}, x_{12}\}$. All model candidates that present an ELPD-LOO difference larger than four and with a difference larger than its standard error were selected for step 3.

3 – We compared the model candidates combining the selected candidates from step 2. If, for example, two different candidates $\{X^{(s1)}, x_{s1+a}\}$ and $\{X^{(s1)}, x_{s1+b}\}$ were selected, we compared the model candidates $\{X^{(s1)}\}$, $\{X^{(s1)}, x_{s1+a}\}$, $\{X^{(s1)}, x_{s1+b}\}$, $\{X^{(s1)}, x_{s1+a}, x_{s1+b}\}$. The model candidate with the least number of predictors and an ELPD-LOO difference >4 as well as a difference larger than the estimated standard error was selected eventually.

We compared all candidate models using leave-one-out cross-validation (LOO-CV) with the Pareto smoothed importance sampling (PSIS-LOO), which is an out-of-sample estimator of predictive model accuracy (Vehtari et al., 2017), implemented in the R package loo (Vehtari et al., 2019)."

Reference: Vehtari, A.: Cross-validation FAQ, https://avehtari.github.io/modelselection/CV-FAQ.html, 2020.

We replaced Table 3 by the new table below, using model "fit1" as reference, and ranking the models by elpd_loo:

Table 3. Comparison of flood-type model candidates of differing complexity and using their expected log pointwise predictive density (ELPD-LOO), ranked by increasing  predictive accuracy, along with differences and their standard errors with reference to model "fit1" (see Table S1 for all model variants).

| Model | ELPD-LOO | ELPD-LOO difference | Standard error of difference |
|---|---|---|---|
| fit1 | 2018.7 | 0.0 | 0.0 |
| fit2 | 2057.3 | 38.6 | 8.7 |
| fit3 | 2093.2 | 74.5 | 12.5 |
| fit4 | 2098.1 | 79.4 | 12.8 |
| fit5 | 2113.4 | 94.7 | 13.6 |
| fit6 | 2124.0 | 105.3 | 14.1 |
| fit7* | 2127.0 | 108.3 | 14.5 |
| fit8* | 2125.4 | 106.8 | 14.5 |
| fit9* | 2126.2 | 107.5 | 14.8 |
| fit10* | 2125.9 | 107.2 | 14.8 |
| fit11 | 2131.8 | 113.1 | 15.1 |
| fit12* | 2134.3 | 115.6 | 15.3 |

* Difference between ELPD-LOO values between two subsequent models is <4.

We replaced Table 4 by the table below, directly comparing the model candidates to model "fit6":

Table 4. Comparison of the flood-type model candidates by their difference in ELPD-LOO using the first six predictors plus one predictor at a time, ranked by  predictive accuracy, along with their differences and the standard error of the differences with reference to model "fit6" (see Table S2 for all model variants)

| Model | ELPD-LOO | ELPD-LOO difference | Standard error of difference |
|---|---|---|---|
| fit8 | 2122.3 | -1.7 | 0.5 |
| fit10 | 2123.2 | -0.9 | 1.4 |
| fit6 | 2124.0 | 0 | 0 |
| fit12 | 2124.2 | 0.2 | 2.0 |
| fit9 | 2124.4 | 0.3 | 2.0 |
| fit7 | 2127.0 | 3.0 | 3.5 |
| fit11 * | 2130.8 | 6.7 | 3.9 |

* model with relevant improvement compared to others (elpd_diff > 4 and elpd_diff > se_diff)

We hope that these revisions now clarify our model selection process and criteria.

***R1-C6. Please provide (in main manuscript) the ELPD differences and SE across the four model types – single level, flood type, event and region for fit6+11 (From SI, table S3). Are there significant differences? Which of the models perform the best?***

A: The table in the supplementary information includes all model variants and is too large for the main manuscript. Hence, we decided to present the information as a chart that complements Table 5. Please note that we addressed the discussion about significant model differences in our reply to the preceding comment.

[Figure]

Figure 1. Comparison of model candidates by their difference in ELPD-LOO using combinations of the first five predictors (fit5) plus predictors 6, 7, and 11, along with their differences and the standard error of the differences with reference to candidate model "fit6" for each model variant.

***R1-C7. Please present some discussion on the model diagnosis (section 3.2)***

A: Thank you for this general suggestion. We added the following paragraph to the Discussion section, following the current line 264.

"After comparing the predictive accuracy estimates of models with different sets of predictors, we selected the model "fit 6+11" that uses water depth, building area, contamination, duration, PLPMs, insurance, and previous flood experience as predictors. Considering that we aim to explore the role of predictors in estimating flood losses, rather than finding the best fit model, chains convergence and posterior predictive checks are a necessary step before interpreting the fitted model (Gabry et al., 2019; Gelman et al., 2020). The three model variants trained with 1,269 datapoints, and sampled with four chains each, converged well, with Gelman-Rubin scales below 1.004 (ideal values are <1.01) and effective sample size ratios above 0.58 (ideal values are >0.5). Visual assessment of the predictive posterior density plot is an important step, whether the model generates data similar to the observed data. Figure 2 shows that the model replicates well the data distribution, and visual inspection confirmed only unimodal estimates. "

Reference: Gabry, J., Simpson, D., Vehtari, A., Betancourt, M., and Gelman, A.: Visualization in Bayesian workflow, Journal of the Royal Statistical Society: Series A (Statistics in Society), 182, 389–402, https://doi.org/10.1111/rssa.12378, 2019.

Gelman, A., Vehtari, A., Simpson, D., Margossian, D., Carpenter, B., Yao, Y., Kennedy, L., Gabry, J., Bürkner, P., Modrák, M.: Bayesian Workflow,

https://statmodeling.stat.columbia.edu/2020/11/10/bayesian-workflow/, last access: 9 December 2020

**R1-C8. The concept of transferability is only discussed in lines 260-263. Please provide further analysis or information regarding how the model addresses the transferability challenge.**

A: Most flood loss models are trained with data of one flood event (or sometimes multiple though largely similar events). Whatever the method, most models will have a single set of parameter estimates for all affected objects.

However, single flood events can affect cities differently across regions, which in turn reflects different socioeconomic and geographic conditions and building codes, for example. These characteristics reflect a given asset's resistance to the hazard process (Thieken et al., 2005) so that a model should have different parameters for different socioeconomic and geographic conditions. These characteristics might not have been explicitly included in the model but may be represented by a proxy such as an administrative region. Thus, we considered a multi-level model structured by regions to explore whether geographic information has to be included in loss modelling.

Furthermore, most similar studies on flood loss estimation consider single flood events to be of a single type only, i.e. fluvial floods, pluvial floods, or coastal floods. We argued, however, that "multiple flood types were reported for the same event, even within the same city, thus giving rise to compound events" (line 43-44): for example, flood waters from river channels that overtop the banks (riverine floods) can meet with runoff-driven flood waters promoted by insufficient urban drainage systems (surface water floods). Thus, a model of a single event should have different parameter estimates for different flood types, and therefore we consider an additional model variant structured by flood types. Finally, we observe that flood preparedness evolved over time, documented, for example, by Kienzler et al. (2015) and Thieken et al. (2016) for Germany. Economic situations may also change the relative value of exposed assets and its recovery or repair costs (Penning-Rowsell, 2005; Kron, 2005). Such effects or evolution in time are challenging to include in loss models, however. Therefore, we considered a third model variant structured by flood events, capturing the timely situation.

We realize now that these structures were presented in the manuscript for flood types, but not in full for the regions and flooding events. We, therefore, add to line 53:

"Here we expand on the model of Mohor et al. (2020) by acknowledging structure in the dataset and explore whether a single regression model can apply not only to different flood types, but also to regions or flooding events. Single flood events can affect cities differently across regions, likely reflecting socioeconomic and geographic conditions and building codes, for example. These characteristics reflect a given asset's resistance to the hazard process (Thieken et al., 2005). These characteristics may differ on the level of administrative regions, and hence we considered a multi-level model variant structured by regions. Additionally, flood preparedness evolved over time, documented, for example, by Kienzler et al. (2015) and

Thieken et al. (2016) for Germany. Economic situations may also change the relative value of exposed assets and its recover or repair costs (Penning-Rowsell, 2005; Kron, 2005). Such changes are challenging to include in loss models, however. Therefore, we considered a third model variant structured by flood events, capturing the timely aspect. Therefore, we estimate relative flood losses in Germany with a Bayesian multilevel model featuring three different groups … "

We argue that exploring these model variants provides more clarity about whether we should use simple average models or more specific multi-level models to be able to transfer predicted loss estimates to new regions, flood types or other structures in the data.

The topic of transferability was also addressed by the second Referee. Therefore, we have added the following paragraph at the end of the Discussion:

> "When addressing transferability, we seek models that can generalize well and go beyond local or case-specific data. Wagenaar et al. (2018) trained two flood loss models using data from two different countries (Germany and the Netherlands) and tested how well each model could predict losses in the other country. They found that the number of flood events in the data was more important than simply the number of reported flood loss cases. Although we trained our models with data from a single country, the data used by Wagenaar et al. (2018) for Germany, comprises six event years across twelve federal states, four river basins (Danube, Rhine, Elbe, and Weser) and four flood types. We expanded on this approach by training models on data from different flood-event years, different flood types, and different regions, thus allowing for a broad range of environmental, administrative, and socio-economic conditions (representing at least Central Europe) that we treat explicitly as grouping levels in our analysis. We argue that exploring these model variants provides more clarity about whether we should use simple average models or more specific multi-level models to be able to transfer predicted loss estimates to new regions, flood types or other structures in the data."

References: Kron, W.: Flood Risk = Hazard • Values • Vulnerability, Water International, 30, 58–68, https://doi.org/10.1080/02508060508691837, 2005.

Penning-Rowsell, E. C.: The benefits of flood and coastal risk management: A handbook of assessment techniques / Edmund Penning-Rowsell … [et al.], Middlesex University Press, London, 89 pp., 2005.

Wagenaar, D., Lüdtke, S., Schröter, K., Bouwer, L. M., & Kreibich, H.: Regional and temporal transferability of multivariable flood damage models, Water Resources Research, 54, 3688–3703. https://doi.org/10.1029/ 2017WR022233, 2018

***R1-C9. SI Table S3 – rephrase Year to Event***

A: The term "Year" was rephrased to "Event".

***R1-C10. Please mention the corresponding SI tables in the respective sections (main manuscript). Where are tables S4 and S5 relevant?***

A: Table S4 and Table S5 show statistical comparison across subsamples of regions and flood events, respectively. Table S5 was already mentioned in line 294 of the original manuscript. Details of Table S4 turned out to be unnecessary for the discussion. Hence, we removed Table S4 and shortened Table S5 to contain only relevant information on hazard characteristics and losses.

On behalf of all co-authors,

Guilherme S. Mohor

---

## Author Comment (AC2) · 26 Feb 2021

We would like to thank the referees for carefully reading our manuscript and for their constructive comments and suggestions. Please find below our point-by-point responses. If a change in the manuscript will be made, we explicitly say so and present the new excerpts below in red.

**Reviewer 2:**

"This paper describes the development of a Bayesian multilevel model for flood damage estimation. These two step models first group observations by event, flood type or region and then build separate models. The study showed that grouping by flood type is most useful for developing transferable flood damage models. The study seems to be carried out well and the writing is generally good."

We thank the referee for taking the time to comment on our manuscript and offering constructive suggestions. Please find below our answers to each point raised.

*R2-C1 One of the main conclusions seems to be that when developing transferable flood damage models it works best to select models by flood type rather than by event or region. This observation is very interesting but this is based on a dataset of just German data. I can imagine that in a more international setting the regional difference might become more important than the flood type differences. I think this needs to be emphasized in the conclusions and I think the paper should therefore more promote the method than the finding (which I expect to be specific to this dataset of a relatively homogenous region).*

A: We agree that our results are informed by the detailed data we have about flood losses in Germany. We emphasised this in our revision (please see below). Yet we point out that the regional variation that our data cover are quite heterogenous. Since urban and land-use planning follows defined administrative and legal guidelines, buildings codes, for example, are constructed differently in different parts of Germany, partly also because of historic reasons. Wagenaar et al. (2018) developed two flood loss models for different countries (Germany and the Netherlands) and tested how well these models could be swapped between countries. They found that the number of flood events in the data was more important than only the number of datapoints from a single event. We expanded on this approach by training models on data from different flood-event years, different flood types, and different regions, thus allowing for a broad range of environmental, administrative, and socio-economic conditions that we treat explicitly as grouping levels in our analysis. The topic of transferability was also addressed by the first Referee. Therefore, we have added the following paragraph at the end of the Discussion:

"When addressing transferability, we seek models that can generalize well and go beyond local or case-specific data. Wagenaar et al. (2018) trained two flood loss models using data from two different countries (Germany and the Netherlands) and tested how well each model could predict losses in the other country. They found that the number of flood events in

the data was more important than simply the number of reported flood loss cases. Although we trained our models with data from a single country, the data used by Wagenaar et al. (2018) for Germany, comprises six event years across twelve federal states, four river basins (Danube, Rhine, Elbe, and Weser) and four flood types. We expanded on this approach by training models on data from different flood-event years, different flood types, and different regions, thus allowing for a broad range of environmental, administrative, and socio-economic conditions (representing at least Central Europe) that we treat explicitly as grouping levels in our analysis. We argue that exploring these model variants provides more clarity about whether we should use simple average models or more specific multi-level models to be able to transfer predicted loss estimates to new regions, flood types or other structures in the data."

***R2-C2 Can you maybe explain better why you go for a multilevel approach rather than just adding variables like flood type, region and event to the dataset? You can then use variable importance to see how much these variables add. In other words can you clarify the added value of this approach better compared to this obvious/simpler alternative approach?***

A: In a previous study by Mohor et al., (2020), we explored with simpler statistical flood-loss models the differences across flood types. We found that slopes and intercepts differed across flood types, while a complete pooling (or average) model had varying intercepts. However, both these approaches overlooked potentially informative structure in the data, for example, the role of flood types, timing, regional characteristics of building codes, or measures of flood preparation. With the multilevel modelling under a Bayesian framework, we trained regression models with varying intersects and varying slopes that duly and explicitly recognise these differing characteristics. One major added value is that the multilevel approach expresses these differing characteristics as individual model components and how they deviate from the average model trained on all the data. The multilevel approach allows us to analyse all data in one model while honouring structure or nominal groups in the data. Thus, the training of the group-specific parameters occurs at the same time so that model parameters can inform each other by means of specified (hyper-)prior distributions. This approach warrants more training data than running stand-alone models on subsets of our data, which in turn are more prone to over- and underfitting and overestimates of the regression coefficients. Given we do have an identifiable structure in our dataset, we see these advantages as welcoming, if not necessary. We extend our presentation of the method explicating these advantages and justifying our method choice, by adding the following to Line 93:

"Bayesian multilevel models weigh the likelihood of observing the given data under the specified model parameters by prior knowledge. Bayesian models thus express the uncertainty in both the prior parameter knowledge and the posterior parameter estimates. The multilevel approach allows us to analyse all data in one model while honouring structure or nominal

groups in the data. Thus, the training of the group-specific parameters occurs at the same time so that model parameters can inform each other by means of specified (hyper-)prior distributions. This approach warrants more training data than running stand-alone models on subsets of our data, which in turn are more prone to over- and underfitting and overestimates of the regression coefficients,  while reducing effects of collinearity, and offering a natural form of penalised regression (McElreath, 2016)."

**R2-C3 In the first sentence of the abstract you note that preparedness is typically ignored. I agree with this statement but its not really what this paper is about and by adding it to the first sentence of the abstract you confuse the reader. So I advice moving this statement.**

A: Thank you for this observation. We agree and changed the abstract accordingly:

"Models for the predictions of monetary losses from floods mainly blend data deemed to represent a single flood type and region. Moreover, these approaches largely ignore indicators of preparedness and how predictors may vary between regions and events, challenging the transferability of flood loss models. We use a flood loss database of [...] "

**R2-C4 Maybe also mention synthetic models in the introduction.**

A: We will reinforce this topic in the introduction. Synthetic models are a good approach to harmonize loss estimation. However, when it comes to including behaviour they are limited by their assumptions. In general, synthetic models tend to reduce (natural) variability of data and are rarely validated (Sairam et al., 2020). We added the following text to the introduction:

"In contrast to empirical models, synthetic models are developed based on expert opinion and offer a good approach to harmonize loss estimations. However, how these models rely on assumptions is problematic when preparedness and other behavioural variables are concerned. In general, synthetic models tend to reduce the variability of data and remain rarely validated (Sairam et al., 2020). Therefore, we train our Bayesian model using reported data. "

Reference: Sairam, N., Schröter, K., Carisi, F., ... & Kreibich, H.: Bayesian Data-Driven approach enhances synthetic flood loss models, Environmental Modelling & Software, 132, 104798. https://doi.org/10.1016/j.envsoft.2020.104798, 2020.

**R2-C5 Line 26: The introduction frames that having a lot of detailed information automatically leads to overfitting and reasons that you therefore need multi-level models. This is not necessarily true, overfitting can be controlled in almost all data-driven methods**

*so its possible to produce more general models with detailed data. Multi-level models are just another way of doing this not the only way.*

A: We argued for a balance between too generalized and too detailed models. We agree that multi-level modelling is not the only way. Indeed, we wrote that "multilevel or hierarchic models offer a compromise […]" (Line 30), meaning that there are of course alternatives. To clarify, we added to this section the importance of other strategies, such as feature selection to minimise overfitting by using cross-validation or regularization (the latter is something which our Bayesian approach offers by design). The revised paragraph now reads:

"In this context, multilevel or hierarchic models are one alternative and offer a compromise between a single pooled model fitted to all data and many different models fitted to subsets of the data sharing a particular attribute or group. Bayesian multilevel models use conditional probability as a basis for learning the model parameters from a weighted compromise between the likelihood of the data being generated by the model and some prior knowledge of the model parameters. These models explicitly account for uncertainty in data, low or imbalanced sample size, and variability of model parameters across different groups (Gelman et al., 2014; McElreath, 2016). There are several approaches to the bias-variance trade-off (McElreath, 2020). We conduct a variable selection through cross-validation to achieve a balance between predictive accuracy and generalization. Using priors in the Bayesian framework is using regularization by design and keeps the model from overfitting the data (McElreath, 2020)."

*R2-C6 The explanation in 2.2.1 and 2.2.2 is a bit difficult to follow. Could you try improve the explanation, maybe using a figure.*

A: Based on the comments of another referee (see R1-C5), we are updating the Tables in section 2.2.2 Model comparison. We added also the following outline of the model selection steps. This new presentation now clarifies our procedure. The following paragraphs replaces lines 156-169:

"On the one hand, testing all models possible without any underlying concept is far from good scientific practice and computationally inefficient; on the other hand, predictors are rarely fully independent. Hence, we fitted candidate models in three steps of model comparison outlined below. We compare the model candidates in each step via the expected log pointwise predictive density (ELPD), which is the sum of a log-probability score of the predictive accuracy for unobserved data. The distribution of these unobserved data is unknown, but we can estimate the predictive accuracy with leave-one-out cross-validation (ELPD-LOO), which is the sum of the log-probability scores for the given data except for one data point at a time (Vehtari et al., 2017; McElreath, 2016). According to Vehtari (2020), an ELPD-LOO difference

>4 may be relevant and should also be compared to the standard error of the difference. Hence, we selected models as follows:

1- We compared models with a gradually increasing number of predictors, based on the prior knowledge of predictor importance reported in a study using single-level linear regression by Mohor et al. (2020). This study considered water depth, for which data are the most widely available and adopted in flood loss models (Gerl et al., 2016) up to a maximum of twelve predictors (Table 1). For example, model 2 (named "fit2") has water depth (WD) and building area (BA) as predictors, while model 3 ("fit3") has the previous two plus contamination (Con) as predictors; model 12 ("fit12") has all twelve predictors (Table 1). The model candidate with an ELPD-LOO difference >4 compared to the previous candidate was selected for the next step.

2 – For the model selected in step 1 – "fit_s1" with predictors $X^{(s1)} = \{x_1, \dots, x_{s1}\}$, we compared models with $X^{(s1)}$ predictors plus one of the remaining predictors at a time, i.e., $\{X^{(s1)}\}$, $\{X^{(s1)}, x_{s1+1}\}$, $\{X^{(s1)}, x_{s1+2}\}$, $\dots$, $\{X^{(s1)}, x_{12}\}$. All model candidates that present an ELPD-LOO difference larger than four and with a difference larger than its standard error were selected for step 3.

3 – We compared the model candidates combining the selected candidates from step 2. If, for example, two different candidates $\{X^{(s1)}, x_{s1+a}\}$ and $\{X^{(s1)}, x_{s1+b}\}$ were selected, we compared the model candidates $\{X^{(s1)}\}$, $\{X^{(s1)}, x_{s1+a}\}$, $\{X^{(s1)}, x_{s1+b}\}$, $\{X^{(s1)}, x_{s1+a}, x_{s1+b}\}$. The model candidate with the least number of predictors and an ELPD-LOO difference >4 as well as a difference larger than the estimated standard error was selected eventually.

We compared all candidate models using leave-one-out cross-validation (LOO-CV) with the Pareto smoothed importance sampling (PSIS-LOO), which is an out-of-sample estimator of predictive model accuracy (Vehtari et al., 2017), implemented in the R package loo (Vehtari et al., 2019). "

***R2-C7 I think the title of 2.2.2 should be more like model tuning rather than model comparison, because you really use the same model but with different settings.***

A: We disagree with this statement. We compare models with different sets of predictors, thus different number of parameters and input data (we maintain the same number of datapoints, but use more predictor variables). Therefore, a better term would be "model selection" and we decided to use this term in the revised version of the paper.

On behalf of all co-authors,

Guilherme S. Mohor